# SoftZoo: A Soft Robot Co-design Benchmark For Locomotion In Diverse Environments

**Tsun-Hsuan Wang**[1,*]**, Pingchuan Ma**[1]**, Andrew Spielberg**[1,4]**, Zhou Xian**[5]**, Hao Zhang**[3]**,**
**Joshua B. Tenenbaum**[1]**, Daniela Rus**[1]**, Chuang Gan**[2,3]
[1]MIT CSAIL, [2]MIT-IBM Watson AI Lab, [3]UMass Amherst, [4]Harvard, [5]CMU
`tsunw@mit.edu, pcma@mit.edu, aespielb@mit.edu, xianz1@andrew.cmu.edu,`
`hao.zhang@umass.edu, jbt@mit.edu, rus@csail.mit.edu, chuangg@umass.edu`

## Abstract

While significant research progress has been made in robot learning for control, unique challenges arise when simultaneously co-optimizing morphology. Existing work has typically been tailored for particular environments or representations. In order to more fully understand inherent design and performance tradeoffs and accelerate the development of new breeds of soft robots, a comprehensive virtual platform — with well-established tasks, environments, and evaluation metrics — is needed. In this work, we introduce SoftZoo, a soft robot co-design platform for locomotion in diverse environments. SoftZoo supports an extensive, naturally-inspired material set, including the ability to simulate environments such as flat ground, desert, wetland, clay, ice, snow, shallow water, and ocean. Further, it provides a variety of tasks relevant for soft robotics, including fast locomotion, agile turning, and path following, as well as differentiable design representations for morphology and control. Combined, these elements form a feature-rich platform for analysis and development of soft robot co-design algorithms. We benchmark prevalent representations and co-design algorithms, and shed light on *1)* the interplay between environment, morphology, and behavior *2)* the importance of design space representations *3)* the ambiguity in muscle formation and controller synthesis and *4)* the value of differentiable physics. We envision that SoftZoo will serve as a standard platform and template an approach toward the development of novel representations and algorithms for co-designing soft robots' behavioral and morphological intelligence. Demos are available on our project page[1].

## 1 Introduction

The natural world demonstrates morphological and behavioral complexity to a degree unexplored in soft robotics. A jellyfish's gently undulating geometry allows it to efficiently travel across large bodies of water; an ostrich's spring-like feet allow for fast, agile motion over widely varying topography; a chameleon's feet allows for dexterous climbing up trees and across branches. Beyond their comparative lack of diversity, soft robots' designs are rarely computationally optimized *in silico* for the environments in which they are to be deployed. The degree of morphological intelligence observed in the natural world would be similarly advantageous in artificial life.

In this paper, we present SoftZoo, a framework for exploring and benchmarking algorithms for co-designing soft robots in behavior and morphology, with emphasis on locomotion tasks. Unlike pure control or physical design optimization, co-design algorithms co-optimize over a robot's brain and body simultaneously, finding more efficacious solutions that exploit their rich interplay (Ma et al., 2021; Spielberg et al., 2021; Bhatia et al., 2021). We have seen examples of integrated morphology and behavior in soft manipulation (Puhlmann et al., 2022), swimming (Katzschmann et al., 2018), flying (Ramezani et al., 2016), and dynamic locomotion (Tang et al., 2020). Each of these robots was designed manually; algorithms that design such robots, and tools for designing the algorithms that design such robots have the potential to accelerate the invention of diverse and capable robots.

---

[*]This work was done during an internship at the MIT-IBM Watson AI Lab. Support for this work was also provided in part by the NSF EFRI Program (Grant No. 1830901), DARPA MCS Program, MIT-IBM Watson AI Lab, and gift funding from MERL, Cisco, and Amazon.

[1]Project Page: `https://sites.google.com/view/softzoo-iclr-2023`

SoftZoo decomposes computational soft robot co-design into four elements: design representations (of morphology and control), tasks for which robots are to be optimized (mathematically, reward or objective functions), environments (including the physical models needed to simulate them), and co-design algorithms. Representationally, SoftZoo provides an unified and flexible interface of robot geometry, body stiffness, and muscle placement that can take operate on common 3D geometric primitives such as point clouds, voxel grids, and meshes. For benchmarking, SoftZoo includes a variety of dynamic tasks important in robotics, such as fast locomotion, agile turning, and path following. To study environmentally-driven robot design and motion, SoftZoo supports an extensive, naturally-inspired material set that allows it to not only simulate hyperelastic soft robots, but also emulate ground, desert, wetland, clay, ice, and snow terrains, as well as shallow and deep bodies of water. SoftZoo provides a differentiable multiphysics engine built atop the material point method (MPM) for simulating these diverse biomes. Differentiability provides a crucial ingredient for the development of co-design algorithms, which increasingly commonly exploit model-based gradients for efficient design search. This focus on differentiable multiphysical environments is in contrast to to previous work (Bhatia et al., 2021; Graule et al., 2022) which relied on simplified physical models with limited phenomena and no differentiability; this limited the types of co-design problems and algorithms to which they could be applied. The combination of differentiable multiphysics simulation with the decomposition of environmentally-driven co-design into its distinct constituent elements (representation, algorithm, environment physics, task) makes SoftZoo particularly well suited to systematically understanding the influence of design representations, physical modeling, and task objectives in the development of soft robot co-design algorithms. In summary, we contribute:

- A soft robot co-design platform for locomotion in diverse environments with the support of an extensive, naturally-inspired material set, a variety of locomotion tasks, an unified interface for robot design, and a differentiable physics engine.
- Algorithmic benchmarks for various representations and learning algorithms, laying a foundation for studying behavioral and morphological intelligence.
- Analysis of *1)* the interplay between environment, morphology, and behavior *2)* the importance of design space representations *3)* the ambiguity in muscle formation and controller synthesis *4)* the value of differentiable physics, with numerical comparisons of gradient-based and gradient-free design algorithms and intelligible examples of where gradient-based co-design fails. This analysis provides insight into the efficacy of different aspects of state-of-the-art methods and steer future co-design algorithm development.

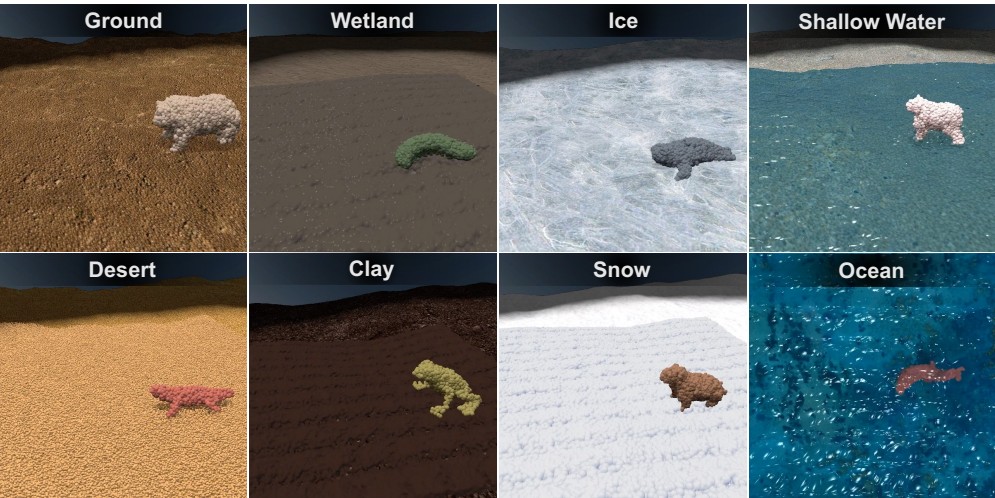

Figure 1: An overview of SoftZoo with demonstration of various biologically-inspired designs.

## 2 SoftZoo

### 2.1 Overview

SoftZoo is a soft robot co-design platform for locomotion in diverse environments. It supports varied, naturally-inspired materials that can construct environments including ground, desert, wetland, clay, ice, shallow water, and ocean. The task set consists of fast locomotion, agile turning, and path

following. The suite provides a seamless and flexible interface of robot design that specifies robot geometry, body stiffness, and muscle placement, and accepts common 3D representations such as point clouds, voxel grids, or meshes. The robot can then be controlled according to muscle groups. The underlying physics engine also supports differentiability that provides model-based gradients, which gains increasing attention in soft robot control and design. In the following, we walk through high-level components of SoftZoo: simulation engine, environment setup, and locomotion tasks.

## 2.2 SIMULATION ENGINE

In this section, we briefly introduce the underlying simulation technique to facilitate later-on discussion. SoftZoo is implemented using the Taichi library (Hu et al., 2019b), which compiles physical simulation code (and its reverse-mode autodifferentiated gradients) for efficient parallelization on GPU hardware. Continuum mechanics in SoftZoo follows the discretization of Moving Least Squares Material Point Method (MLS-MPM) (Hu et al., 2018), a variant of B-spline MPM (Stomakhin et al., 2013) with improved efficiency. Multimaterial simulation and signed-distance-function-based boundary conditions for terrain are implemented to construct diverse environments.

**Material Point Method (MPM).** The material point method is a hybrid Eulerian-Lagrangian method, where both Lagrangian particles and Eulerian grid nodes are used to transfer simulation state information back-and-forth. At a high level, MPM is comprised of three major steps: particle-to-grid transfer (P2G), grid operation, and grid-to-particle transfer (G2P). Material properties, including position $\mathbf{x}_p$, velocity $\mathbf{v}_p$, mass $m_p$, deformation gradients $\mathbf{F}_p$, and affine velocity field $\mathbf{C}_p$, are stored in Lagrangian particles that move through space and time. MPM allows large deformation, automatic treatment of self-collision and fracture, and simple realization of multi-material interaction, and is hence well-suited for soft robots in diverse environments. We show the governing equations used by MPM and more implementation details in Appendix D.

**Contact Model.** The terrain in SoftZoo is represented as meshes that interact with either the robot body or other ground cover materials such as snow or sand. We employ a grid-based contact treatment (Stomakhin et al., 2013) to handle particle terrain-particle collision, using a Coulomb friction model. We compute the surface normal of the terrain to measure the signed distance function (SDF) and construct boundary conditions for velocity in grid space.

**Multiphysical Materials.** We present a set of environments with multiphysical material support. The materials cover a diverse set of physical phenomena, including hyperelasticity, plasticity, fluidity, and inter-particle friction. These phenomena combine to model common real-world materials such as sand, snow, rubber, mud, water, and more. We list all environments with their corresponding multiphysical materials in Appendix E. In addition to categorical choices of environments, we also provide a set of elastic constitutive models (*e.g.*, St. Venant-Kirchhoff, corotated linear, neo-Hookean, etc.) and expose their parameters so that the practitioners can easily fine-tune the material behaviors. In our experiments, we use neo-Hookean material since it *a)* is known to be accurate for modeling silicone-rubber-like materials common in real-world soft robots, *b)* skips the costly calculation of singular value decomposition (SVD) and *c)* results in an improvement in simulation speed and numerical stability and reduces gradient instability caused by singular value degeneracy.

**Actuation Model.** We define our actuation model as an anisotropic muscle installed along specified particles. Each muscular element generates a directional force along a unit fiber vector $\mathbf{f}$. We realize this actuation model using the anisotropic elastic energy from Ma et al. (2021) $\Psi = s\|l - a\|^2$, where $l = \|\mathbf{F}\mathbf{f}\|^2$, $\mathbf{F}$ is the deformation gradient, $s$ is a muscular stiffness parameter, and $a$ is the actuation signal provided by a controller. This energy is added to the constitutive material energy of particles along which the muscle fiber is installed.

## 2.3 ENVIRONMENT SETUP

**Initialization.** Environment construction requires three steps: terrain generation, robot placement, and material covering. First, we procedurally generate terrain height-maps using Perlin noise with user-defined height range and other parameters for roughness. The generated height map is converted to a mesh for ground surface. We then instantiate a robot using specified design parameters. A user may choose to have the robot placed randomly or at a fixed pre-defined position on the terrain. Then, we compute an occupancy map in the Eulerian grid in MPM based on particles of robot body and terrain's SDF. Finally, we use the terrain's surface normal to layer particles of specified terrain materials atop it, using the occupancy map to avoid particle placement in non-free space.

Table 1: Large-scale benchmark of biologically-inspired designs in SoftZoo. Each entry shows results from differentiable physics (left) and RL (right). The higher the better.

| Task | Animal | Ground | Ice | Wetland | Environment Clay | Desert | Snow | Ocean |
|---|---|---|---|---|---|---|---|---|
| Movement Speed | Baby Seal | 0.122 / **0.154** | **0.048** / 0.010 | 0.032 / 0.020 | 0.012 / 0.005 | 0.059 / 0.034 | 0.039 / 0.016 | 0.033 / 0.029 |
| | Caterpillar | 0.080 / 0.032 | 0.023 / 0.006 | **0.052** / 0.016 | 0.032 / 0.015 | 0.053 / 0.032 | 0.047 / 0.017 | 0.134 / **0.181** |
| | Fish | 0.053 / 0.033 | 0.029 / 0.011 | 0.026 / 0.013 | **0.037** / 0.014 | **0.115** / 0.022 | **0.087** / 0.042 | 0.084 / 0.151 |
| | Panda | 0.046 / 0.019 | 0.038 / 0.006 | 0.016 / 0.008 | 0.019 / 0.005 | 0.023 / 0.009 | 0.031 / 0.004 | 0.024 / 0.007 |
| Turning | Baby Seal | 0.067 / **0.077** | **0.058** / 0.024 | 0.014 / 0.008 | 0.021 / 0.011 | **0.051** / 0.026 | 0.047 / 0.028 | 0.059 / 0.020 |
| | Caterpillar | 0.053 / 0.021 | 0.021 / 0.009 | **0.040** / 0.006 | **0.027** / 0.005 | 0.034 / 0.015 | **0.069** / 0.006 | 0.195 / **0.358** |
| | Fish | 0.032 / 0.047 | 0.023 / 0.012 | 0.023 / 0.010 | 0.015 / 0.007 | 0.021 / 0.019 | 0.028 / 0.029 | 0.041 / 0.013 |
| | Panda | 0.064 / 0.014 | 0.023 / 0.003 | 0.009 / 0.004 | 0.008 / 0.001 | 0.014 / 0.003 | 0.013 / 0.002 | 0.035 / 0.031 |
| Velocity Tracking | Baby Seal | 0.343 / 0.410 | 0.257 / 0.194 | 0.249 / 0.222 | 0.231 / 0.205 | 0.290 / 0.265 | 0.215 / 0.198 | 0.379 / 0.068 |
| | Caterpillar | 0.502 / 0.368 | 0.101 / 0.156 | 0.426 / 0.192 | 0.282 / 0.058 | 0.441 / 0.376 | 0.379 / 0.133 | 0.555 / **0.714** |
| | Fish | 0.216 / 0.256 | 0.416 / 0.319 | 0.221 / 0.236 | 0.234 / **0.303** | **0.457** / 0.311 | **0.638** / 0.307 | 0.657 / 0.574 |
| | Panda | **0.575** / 0.424 | **0.555** / 0.383 | **0.536** / 0.534 | 0.153 / 0.195 | 0.450 / 0.359 | 0.472 / 0.288 | 0.395 / 0.220 |
| Waypoint Following | Baby Seal | -0.012 / -0.014 | -0.018 / -0.027 | -0.018 / -0.027 | **-0.020** / -0.026 | -0.012 / -0.025 | -0.014 / -0.026 | -0.010 / -0.026 |
| | Caterpillar | -0.013 / -0.016 | -0.019 / -0.028 | **-0.016** / -0.028 | **-0.020** / -0.027 | -0.013 / -0.027 | -0.018 / -0.026 | **-0.002** / -0.019 |
| | Fish | -0.004 / -0.016 | -0.016 / -0.026 | -0.022 / -0.029 | -0.024 / -0.027 | **-0.007** / -0.024 | **-0.003** / -0.024 | -0.005 / -0.023 |
| | Panda | **-0.003** / -0.014 | **-0.014** / -0.025 | -0.021 / -0.027 | -0.023 / -0.028 | -0.010 / -0.024 | -0.008 / -0.025 | -0.013 / -0.026 |

A co-design algorithm consists of *1)* a design optimizer that proposes a robot design at the start of a simulation trial (an "episode" in reinforcement learning terminology) and *2)* a control optimizer that specifies a controller that determines robot actuation based on its observed state. Accordingly, each task interfaces with the algorithm through the *robot design interface*, *observation*, *action*, and *reward*. We introduce each element as follows.

**Robot Design Interface.** Robot design involves specification of geometry, stiffness, and muscle placement. We integrate these design specifications into Lagrangian particles in MPM simulation. Geometry is modeled by mass $m_p \in \mathbb{R}$ (clamping regions of sufficiently low mass to $0$); stiffness is modeled by a the Young's modulus in elastic material $s_p \in \mathbb{R}$. We remove non-existing (zero-mass) particles to eliminate their effect. (Low and zero mass particles cause numerical instabilities during simulation.) For muscle placement $\mathbf{r}_p \in \mathbb{R}^K$ on a robot with at most $K$ actuators, we augment each actuated particle with a $K$-dimensional unit-vector where the magnitude of component $i$ specifies the contribution of actuator $i$ to that particle as well as a 3-dimensional unit-vector to specify muscle direction $\mathbf{f}_p \in \mathbb{R}^3$. While a robot is represented as a particle set in the simulator, SoftZoo allows common 3D representations other than point clouds such as voxel grids or meshes. We instantiate a set of particles in a bounding base shape (*e.g.*, a box or ellipsoid), which defines the workspace of a robot design with position $\{\mathbf{x}_p\}$, velocity $\{\mathbf{v}_p\}$, and other attributes required in MPM. For voxel grids, we utilize a voxelizer to transfer voxel occupancy to particle mass that resides in that voxel. For meshes, we compute SDF for particles in the bounding base shape and assign non-zero mass to sub-zero level set (those with negative signed distance).

**Observation.** Robot state observations, which are fed to controllers, are computed at every step in an episode, consisting of robot proprioceptive information, environmental information, and task-related information. For robot proprioceptive information, considering states of all particles in robot body is unrealistic from both the perspective of sensor placement and computational tractability. Instead, we compute the position and velocity centroids of the robot body, or of pre-defined body parts. The environment is summarized as a *semantic occupancy map*; from the MPM Eulerian grid, a 3D voxel grid is constructed in with each voxel indicates occupancy of terrain (below the SDF boundary), terrain material particles, or the robot. Finally, task-related information provides sufficient specifications in order to solve certain task, *e.g.*, target waypoints to be followed.

**Action.** At each time step, the simulator queries the robot controller for an action vector $\mathbf{u} \in \mathbb{R}^K$. (Note that the time step is at the time scale of a robot controller, different from simulation steps often referred to as substeps.) We use the same action vector for all simulation steps within an environment time step. The returned action vector is of length as $K$, the maximal number of actuators. The action space bound is set to achieve reasonable robot motion without easily causing numerical fracture of robot body. We use $[-0.3, 0.3]$ as coordinate-wise bound in this paper.

**Reward.** A robot's task performance is quantitatively measured by a scalar-valued reward function. The reward definition is task-specific and computed at every time step to provide feedback signal to the robot. Please refer to Section 2.4 for detailed descriptions of reward for every tasks.

## 2.4 LOCOMOTION TASKS

We only mention the high-level goal of each task. Please refer to Section B for more details.

| Fish Forward in Desert | Caterpillar Forward in Wetland | Panda Turning on Ground | Baby Seal Turning on Ice |
|---|---|---|---|

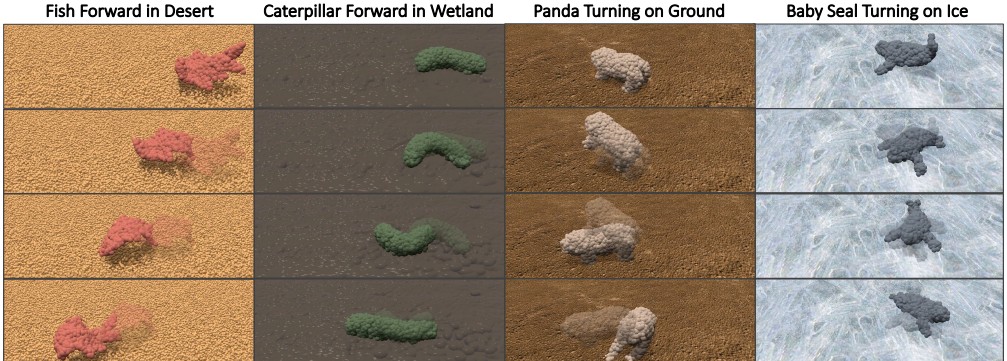

Figure 2: Unique motions arising from morphology and environment to achieve locomotion.

**Movement Speed.** In this task, the robot aims to move as fast as possible along a target direction.

**Turning.** In this task, the robot is encouraged to turn as fast as possible counter-clockwise about the upward direction of robot's canonical pose.

**Velocity Tracking.** In this task, the robot is required to track a series of timestamped velocities.

**Waypoint Following.** In this task, the robot needs to follow a sequence of waypoints.

## 3 EXPERIMENTS

### 3.1 BIOLOGICALLY-INSPIRED MORPHOLOGIES

To study the interplay between environment, morphology, and behavior, we first use a set of animal meshes (selected from https://www.davidoreilly.com/library) as robot bodies and label muscle locations. Through optimizing their controllers, we observe that different designs exhibit better-performing behavior in different environments.

**Muscle Annotation.** We implement a semi-automatic muscle annotator that first converts a mesh into a point cloud, secondly performs K-means clustering with user-defined number of body groups, and finally applies principal component analysis on indepdendent clusters to extract muscle direction. Users may then fine-tune the resulting muscle placement.

**Animals.** We test four animals: *Baby Seal*, *Caterpillar*, *Fish*, and *Panda*. These animals are chosen since they exhibit distinct strategies in nature. We refer the readers to Section K for visualization of each animal-inspired design, including their muscle layouts, and detailed description.

**Large-scale Benchmarking.** Table 1 shows the performance of the four animals when optimized for each locomotion task, in each environment. We parameterize the controller based on a set of sine function bases with different frequencies, phases, and biases (see Section H). We perform control optimization with differentiable physics and RL (Schulman et al., 2017), corresponding to the left/right values in every entry respectively. We would like to emphasize that this experiment is not meant to draw a conclusion on superiority of differentiable physics or RL. Rather, we highlight interesting emergent strategies of different morphologies in response to diverse environments. A We find that the *Baby Seal* morphology is particularly well-suited for *Ground* and *Ice*, the *Caterpillar* morphology is well-suited for *Wetland* and *Clay*, the *Fish* morphology is well-suited for *Desert* and *Snow*, and both the *Caterpillar* and *Fish* morphologies are well-suited for the *Ocean*.

**Environmentally- and Morphologically-driven Motion.** Based on previous quantitative analysis, in Figure 2, we showcase several interesting emergent motion of different animals. The *Fish* surprisingly demonstrates efficient movement on granular materials (*Desert*), by lifting its torso to reduce friction and pushing itself forward with its tail. The *Caterpillar* produces a left-to-right undulation motion to locomote in *Wetland*; the strategy allows it to disperse mud as it moves forward. The *Panda*'s legs act in a spring-like fashion to perform galloping motion on flat terrain. Finally, the *Baby Seal* uses its tail muscle to create a flapping motion to move on the low-friction *Ice* terrain, where push-off would otherwise be difficult.

Table 2: Quantitative comparison of design space representations.

| Representation | Performance |
| --- | --- |
| Particle | 0.027 |
| Voxel | 0.023 |
| Implicit Function | 0.113 |
| Diff-CPPN | 0.091 |
| SDF-Lerp | 0.152 |
| Wass-Barycenter | 0.158 |

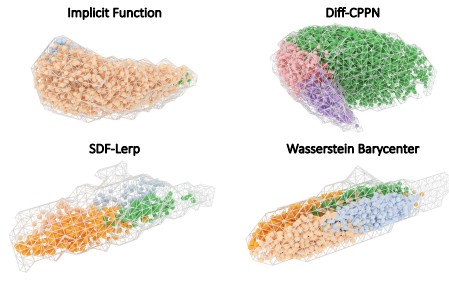

Figure 3: Visualization of optimized designs.

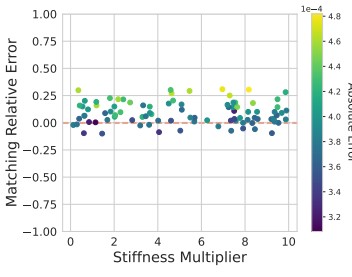

(a) Under different stiffness.

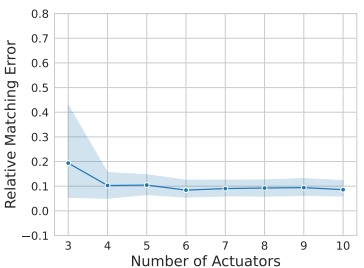

(b) Under different muscle placement.

Figure 4: Ambiguity of muscle formation and controller synthesis.

## 3.2 THE IMPORTANCE OF DESIGN SPACE REPRESENTATION

In this section, we demonstrate the importance of the design space representation for morphology optimization. We apply gradient-based optimization methods across a wide variety of morphological design representations fixing the controller. We focus on achieving fast movement speed in the *Ocean* environment since aquatic creatures tend to manifest a more canonical muscle distribution, typically antagonistic muscle pairs along each side of the body. The low failure rate of this task means that we achieve a wide spread of performances, determined by the design representation, which allows us to compare the effect of each on performance.

**Baselines.** We implement a variety of design space representations. We briefly mention the high-level concept of each method as follows. Please refer to Section I for more in-depth description.

*Particle Representation.* The geometry, stiffness, and muscle placement is directly specified at the particle level, with each particle possessing its own distinct parameterization.

*Voxel Representation.* This is similar to particle representation but specified at the voxel level.

*Implicit Function (Mescheder et al., 2019).* We use a shared multi-layer perceptron that takes in particle coordinates and outputs the design specification for the corresponding particle.

*Diff-CPPN (Fernando et al., 2016).* We adapt the differentiable CPPN, which provides a global mapping that converts particle coordinates to a design specification.

*SDF-Lerp.* Given a set of design primitives (with design specification obtained by using the technique in Section 3.1), we compute SDF based on each design primitives for particles in robot design workspace. For each particle, we then linearly interpolate the signed distances and set occupancy for those with negative values to obtain robot geometry. We directly perform linear interpolation on stiffness and muscle placement of the primitive set. For muscle direction, we use weighted rotation averaging with special orthogonal Procrustes orthonormalization (Brégier, 2021).

*Wass-Barycenter (Ma et al., 2021).* We compute Wasserstein barycenter coordinates based on a set of coefficients to obtain robot geometry. We follow *SDF-Lerp* for stiffness and muscle placement.

**Design Optimization.** In Table 2, we show the results of design optimization. We can roughly categorize these design space representations into (1) no structural prior (*Particle* and *Voxel*) (2) preference for spatial smoothness (*Implicit Function* and *Diff-CPPN*) and (3) highly-structured design basis (*SDF-Lerp* and *Wass-Barycenter*). We observe superior performance as increasing inductive prior is injected into the representation. The use of design primitives casts the problem to composition of existing functional building blocks and greatly reduces the search space, leading to

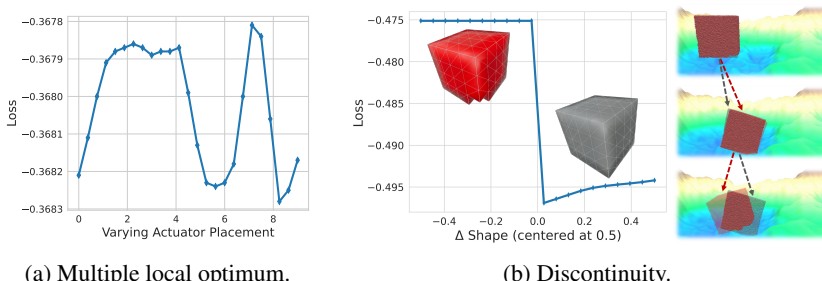

(a) Multiple local optimum.            (b) Discontinuity.

Figure 5: Examples of pathological loss landscape for design optimization.

effective optimization. Further, structural priors, lead to designs with fewer thin or sharply changing features; such extreme designs have unwanted artifacts such as numerical fracture during motion. Figure 3 presents optimized designs: meshes indicate geometry and colored point clouds indicate active muscle groups. These results suggest the development of novel representation for robot design that facilitates improved optimization and learning.

### 3.3 AMBIGUITY BETWEEN MUSCLE STIFFNESS AND ACTUATION SIGNAL STRENGTH

In this section, we investigate the ambiguity between muscle formation and actuation from the controller. We aim to discover if there exists unidentifiability between design and control optimization. We adopt trajectory optimization for control in this set of experiments since it has the best flexibility.

**Stiffness as "Static" Actuation.** Here, we explore whether optimizing active actuation can reproduce motion of robots with different static muscle stiffness. With a fixed robot geometry and muscle placement, we randomly sample 100 sets of controllers and muscle stiffness, and record trajectories of all particles for 100 frames. We then try to fit a controller for robots with a different set of stiffness and measure how well they can match the last frame of the pre-collected dataset. We use the Earth Mover's Distance (EMD) loss (Achlioptas et al., 2018) and differentiable physics for training. Specifically, we use EMD to compare the difference between the motions of each particle set at the end of each trajectory. In Figure 4a, we show matching error distribution across different stiffness multipliers (with respect to the original stiffness of the robot). This experiment indicates we can roughly replicate the motion of robot with different stiffness by control optimization. This aligns with the muscle model, where we can interpret stiffness as a static component of actuation.

**Approximating Motion of Random Muscle Placement.** Here, we investigate if we can reproduce the motions of robots with different muscle placements by optimizing their controllers. With fixed robot geometry and stiffness, we randomly sample 6 sets of controllers and muscle placements with 3 actuators. We follow the same training procedure mentioned previously to fit controllers for robots with a different set of muscle placement. In Figure 4b, we show relative matching error at different number of actuators with shaded area as confidence interval from the 6 random seeds. Note that we adopt a fixed muscle direction and soft muscle placement for simplicity. The plot suggests that with redundant ($> 3$ in this case) actuators, robot motion can be roughly reproduced with control optimization even under different muscle placement.

Overall, we demonstrate the ambiguity between muscle construction and actuation that may induce challenges in co-design optimization. However, these results also unveil the potential of casting muscle optimization to control optimization in soft robot co-design.

### 3.4 THE GOOD AND BAD OF DIFFERENTIABLE PHYSICS FOR SOFT ROBOT CO-DESIGN

**The Value of Differentiable Physics.** Gradient-based control optimization methods powered by differentiable simulation, are increasingly popular in computational soft robotics. Such optimizers can efficiently search for optimal solutions and decrease the number of computationally-intensive simulation episodes needed to achieve optimal results in various computational soft robotics problems compared to model-free approaches such as evolutionary strategies or RL. Despite its effectiveness, the local nature of gradient-based methods poses issues for optimization. While recent research has rigorously studied differentiable physics in control (Suh et al., 2022), similar investigation is lacking in the context of design. Here, we take a preliminary step to analyze gradient for design optimization. In Figure 5a, we show the loss landscape of a single parameter of actuator placement. We consider a voxel grid and smoothly (with soft actuator placement) transition the membership of the

Table 3: The efficacy of co-optimization.

| Optimization Target | Performance |
|---|---|
| Controller | 0.107 |
| Design | 0.152 |
| Design + Control | 0.332 |

Table 4: Co-design with gradient-free/based methods.

| Method | Performance |
|---|---|
| Evolution Strategy | 0.247 |
| Diff-physics | 0.332 |

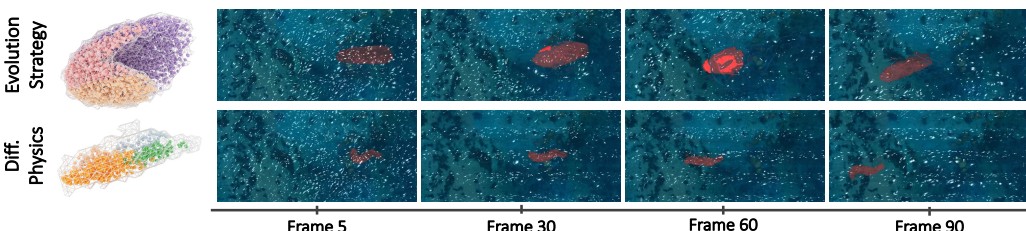

Figure 6: Swimming motion sequences of co-design baselines.

actuator in a voxel. We can observe multiple local optima, which may trap gradient-based optimization. In Figure 5b, we show the discontinuity of loss resulted from shape changes. A box with a missing corner (red) ends up at a very different location from that of a complete box (gray), as it topples when placed in metastable configurations. These results present unique yet understandable challenges for differentiable physics in robot design, which suggests more future research to better leverage gradient information.

## 3.5 CO-OPTIMIZING SOFT ROBOTIC SWIMMERS

In this section, we demonstrate co-design results for a swimmer in the *Ocean* environment. First, in Table 3, we compare between (1) optimizing control only (2) optimizing design only (3) co-optimizing design and control. We use *SDF-Lerp* as design space representation and differentiable physics for optimization. To generate meaningful results, for the control-only case, we use a fish-like design; and for the design-only case, we use a controller trained for a fish not included in the design primitive of *SDF-Lerp*. The outcome verifies the effectiveness of generating superior soft robotic swimmer with co-design. In Table 4, we compare co-design results from evolution strategy (ES) and differentiable physics. We follow the above-mentioned setup for differentiable physics. For evolution strategy, we adopt a common baseline: CPPN representation (Stanley, 2007) with HyperNEAT (Stanley et al., 2009); to make control optimization consistent with evolution strategy, we use CMA-ES (Hansen et al., 2003). Furthermore, we show the optimized design and the corresponding motion sequence of both methods in Figure 6. Interestingly, the better-performing co-design result much resembles the results of performing design optimization only, shown in Figure 3.

## 4 RELATED WORK

**Co-Design of Soft Bodied Machines.** Evolutionary algorithms (EAs) have been used to design virtual agents since the pioneering work of Sims (1994); as these algorithms improved the problems they could be applied to grew in complexity. Cheney et al. (2014b) demonstrated EAs for co-designing soft robots over shape, materials, and actuation for open-loop cyclic controllers; follow-on work explored soft robots with circuitry (Cheney et al., 2014a) and those grown from virtual cells (Joachimczak et al., 2016). Evolutionarily designed soft robots were demonstrated to be physically manufacturable using soft foams (Hiller & Lipson, 2011), and biological cells (Kriegman et al., 2020). Virtually all such results have been powered by the neuroevolution of augmenting topologies (NEAT) algorithm (Stanley & Miikkulainen, 2002), and, typically, compositional pattern producing networks (CPPN) (Stanley, 2007) for morphological design representation. Recent work has explored other effectual heuristic search algorithms, such as simulated annealing, for optimizing locomoting soft robots represented by grammars (Van Diepen & Shea, 2019; van Diepen & Shea, 2022). Recently, Bhatia et al. (2021) provided an expansive suite of benchmark tasks for evaluating algorithms for co-designing soft robots with closed-loop controllers. That work provides a baseline co-design method that employs CPPN-NEAT for morphological search and reinforcement learning for control optimization (Schulman et al., 2017). Similar to (Bhatia et al., 2021), our work benchmarks EAs alongside competing methods. We provide a suite of environmentally-diverse tasks in a

differentiable simulation environment, which allows the use of efficient gradient-based search algorithms. Further, we focus on a systematic decoupling of design space representation, control, task and environment, and search algorithms in order to distill the influence of each.

**Differentiable Simulation for Soft Robot Optimization.** A differentiable simulator is one in which the gradient of any measurement of the system can be analytically computed with respect to any variable of the system, which can include behavioral (control) and physical (morphological and environmental) parameters. Differentiability provides particular value for soft robotics; gradient-based optimization algorithms can reduce the number of (typically expensive) simulations needed to solve computational control and design problems. Differentiable material point method simulation (MPM) has been used in co-optimizing soft robots over closed-loop controllers and spatially varying material parameters (Hu et al., 2019c; Spielberg et al., 2021), as well as proprioceptive models Spielberg et al. (2019). Co-optimization procedures rooted in differentiable simulation have also been applied to more traditional finite element representations of soft robots. Notably, (Ma et al., 2021) demonstrated efficient co-optimization of swimming soft robots' geometry, actuators, and control, incorporating a learned (differentiable) Wasserstein basis for tractable search over high-dimensional morphological design spaces. Follow-on work further showed that differentiable simulators can be combined with learned dynamics for co-design (Nava et al., 2022). Such finite-element-based representations, however, have struggled with smoothly differentiating through rich contact.

Our work presents a rich comparison of differentiable and non-differentiable approaches for soft-bodied co-design, as well as a set of baseline methods. We present a differentiable soft-bodied simulation environment capable of modeling multiphysical materials. Our environment is based in MPM, because of its ability to naturally and differentiably handle contact and multiphysical coupling. Similar to other recent work, such as Suh et al. (2022) and Huang et al. (2021), we perform a thorough analysis (in our case, empirical) of the value and limitations of differentiable simulation environments, with an emphasis on cyberphysical co-design.

**Environmentally-Driven Computational Agent Design.** Similar to their biological counterparts, virtual creatures have different optimized forms depending on their environment. Auerbach & Bongard (2014) and Corucci et al. (2017) analyzed real-world biological morphological features from the perspective of EAs; Cheney et al. (2015) studied the interplay between of virtual terrain and robot geometry for soft-bodied locomotion locomotion. In rigid robotics where simulation is less computationally expensive, online data-driven neural generative design methods are emerging. These include methods that reason over parameterized geometries Ha (2019), topological structure Zhao et al. (2020); Xu et al. (2021); Hu et al. (2022), and shapeshifting behavior Pathak et al. (2019). These methods currently require too much simulation-based data generation for soft robotics applications, but are useful as inspiration for future research directions.

## 5 CONCLUSION

In this work, we introduced SoftZoo, a soft robot co-design platform for locomotion in diverse environments. By using SoftZoo to investigate the interplay between design representations, environments, tasks, and co-design algorithms, we have found that *1)* emergent motions are driven by environmental and morphological factors in a way that often mirrors nature *2)* injecting design priors in design space representation can improve optimization effectiveness *3)* muscle stiffness and controllers introduce redudancy in design spaces, and *4)* trapping local minima are common even in very simple morphological design problems. **Impact and Future Direction.** SoftZoo provides a well-established platform that allows for systematic training and evaluation of soft robot co-design algorithms; we hope it will accelerate algorithmic co-design development. It also lays a cornerstone for studying morphological and behavioral intelligence under diverse environments. The extensive experiments conducted using SoftZoo not only sheds light on the significance of co-design but also identifies concrete challenges from various perspectives. Based on our findings, we suggest several future directions: *1)* 3D representation learning to construct more effective and flexible design space representations *2)* morphology-aware policy learning as an alternative formulation that more elegantly handles the inter-dependency between design and control optimization *3)* principled approaches to combine differentiable physics and gradient-free methods like RL or EAs that marries priors from physics to optimization techniques less susceptible to pathological loss landscapes. Overall, we believe SoftZoo paves a road to study morphological and behavioral intelligence, and bridges soft robot co-design with a variety of related research topics.

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

## A NOTATIONS

Table A shows a list of notations used in the description of SoftZoo, where the subscript $p$ indicates association with particles. Note that actuation is defined per particle and action vector is defined as the output of controllers (which can be very low-dimensional such as $\mathbb{R}^{10}$), i.e., $a_p = \mathbf{u} \cdot \mathbf{f}_p$.

| Symbol | Type | Description |
|--------|------|-------------|
| $m_p$ | scalar | mass |
| $s_p$ | scalar | stiffness |
| $a_p$ | scalar | actuation |
| $\mathbf{u}$ | vector | action vector |
| $\mathbf{x}_p$ | vector | position |
| $\mathbf{v}_p$ | vector | velocity |
| $\mathbf{r}_p$ | vector | muscle placement / membership |
| $\mathbf{f}_p$ | vector | muscle direction |
| $\mathbf{F}_p$ | matrix | deformation gradient |
| $\boldsymbol{\sigma}_p$ | matrix | Cauchy stress |
| $C_p$ | matrix | affine velocity field |

## B LOCOMOTION TASKS

### B.1 MOVEMENT SPEED

In this task, the goal of the robot is to move as fast as possible along a pre-defined direction. We compute the average velocity of all existing (non-zero-mass) particles on robot body and project it to the target direction to obtain a scalar estimate. We thus write the metric per timestep as,

$$r_{\text{speed}} = \sum_p m_p (\mathbf{v}_p \cdot \mathbf{t})$$

where $m_p$ is mass, $\mathbf{v}_p$ is velocity, and $\mathbf{t}$ is a pre-defined target direction.

### B.2 TURNING

In this task, the robot is encouraged to turn as fast as possible counter-clockwise about the upward direction of robot's canonical pose. We first compute the relative position of all particles on robot body with respect to its center of mass. We then compute the cross product of the upward direction followed by unit-vector normalization to obtain tangential directions for all particle while turning. Finally, we compute velocity projection of every particle along the tangent and return the average as the measure of turning performance. We thus write the metric per timestep as,

$$r_{\text{turning}} = \sum_p m_p (\mathbf{v}_p \cdot \frac{\mathbf{d} \times \mathbf{v}_p}{||\mathbf{d} \times \mathbf{v}_p||})$$

where $m_p$ is mass, $\mathbf{v}_p$ is velocity, and $\mathbf{d}$ is the up direction (defined manually based on robot canonical forward direction).

### B.3 VELOCITY TRACKING

In this task, the robot is required to track a series of timestamped velocities. We use quintic polynomials formulated as a function of time for path parameterization since it generates smooth trajectories with easy access to different orders of derivative. A path can be fully specified with start and target position $\mathbf{x}_s, \mathbf{x}_t$, velocity $\mathbf{v}_s$, $\mathbf{v}_t$, and acceleration $\mathbf{a}_s$, $\mathbf{a}_t$. We can then obtain the quintic polynomial as,

$$\mathbf{x}_{\text{path}}(t) = \mathbf{c}_0 + \mathbf{c}_1 t + \mathbf{c}_2 t^2 + \mathbf{c}_3 t^3 + \mathbf{c}_4 t^4 + \mathbf{c}_5 t^5$$

where $t$ is time and $\mathbf{c}_0, \mathbf{c}_1, \mathbf{c}_2, \mathbf{c}_3, \mathbf{c}_4, \mathbf{c}_5 \in \mathbb{R}^3$ are coefficients. After setting a target state, we query the first-order derivative of the polynomial at each environment time step as the target velocity,

$$\mathbf{v}_{\text{path}}(t) = \mathbf{c}_1 + 2\mathbf{c}_2 t + 3\mathbf{c}_3 t^2 + 4\mathbf{c}_4 t^3 + 5\mathbf{c}_5 t^4$$

We write target velocity at a time instance as $\mathbf{v}_{\text{target}}$ for the following. Furthermore, due to the deforming nature of soft robot, we need to estimate the heading of the robot in order compare it with the target velocity. We manually label particles corresponding to a head and a tail of the robot *a priori* $p_{\text{head}}, p_{\text{tail}}$ based on its canonical heading direction.

$$\mathbf{d} = \frac{\mathbf{x}_{p_{\text{head}}} - \mathbf{x}_{p_{\text{tail}}}}{||\mathbf{x}_{p_{\text{head}}} - \mathbf{x}_{p_{\text{tail}}}||}$$

During robot motion, we extract rotation from deformation gradient of these particles and inversely transform them back to material space to compute heading direction.

$$\mathbf{F}_p = \mathbf{U}_p \mathbf{R}_p^T$$
$$\mathbf{v}_p^{\text{proj}} = (\mathbf{R}_p^T \mathbf{v}_p \cdot \mathbf{d})\mathbf{d}$$

where $\mathbf{F}_p$ is deformation gradient, $\mathbf{U}_p, \mathbf{R}_p^T$ are the results of polar decomposition. Velocities of all particles on robot body are then projected to the heading direction and averaged in order to compare to the target velocity. We separate the measurement of magnitude and direction as this allows different weighting of the two terms. We can then write the metric per timestep as,

$$r_{\text{vel-track}} = \alpha_{\text{mag}} r_{\text{mag},p} + \alpha_{\text{dir}} r_{\text{dir},p},$$
$$\text{where } r_{\text{mag},p} = -(||\mathbf{v}_{\text{target}}|| - ||\bar{\mathbf{v}}^{\text{proj}}||)^2$$
$$r_{\text{dir},p} = \mathbf{v}_{\text{target}} \cdot \frac{\bar{\mathbf{v}}^{\text{proj}}}{||\bar{\mathbf{v}}^{\text{proj}}||}$$
$$\bar{\mathbf{v}}^{\text{proj}} = \sum_p m_p \mathbf{v}_p^{\text{proj}}$$

where $\alpha_{\text{mag}}, \alpha_{\text{dir}}$ are coefficients of magnitude and direction term respectively and we set $\alpha_{\text{mag}} = 0.1$ and $\alpha_{\text{dir}} = 0.9$. We put more emphasis on the alignment of direction since it better indicates maneuverability.

### B.4 WAYPOINT FOLLOWING

In this task, the robot needs to follow a sequence of waypoints. The waypoints are generated by the above-mentioned quintic polynomial method $\mathbf{x}_{\text{path}}(t)$. We compute root mean squared error between robot center of mass and the target waypoint for each time step. We denote target position at a time instance as $\mathbf{x}_{\text{target}}$. We can then write the metric per timestep as,

$$r_{\text{waypoint}} = (\mathbf{x}_{\text{target}} - \sum_p m_p \mathbf{x}_p)^2$$

Remark that well-performing velocity tracking can induce large waypoint following error but trace out similar-shaped yet different-scaled trajectories. Both serve a role as distinct aspects of evaluating path following.

## C PLATFORM COMPARISON

A comprehensive comparison to existing soft robot platform is shown in Table 5.

## D CONTINUUM MECHANICS SIMULATION

We formulate the continuum mechanics simulation in the framework of the moving least squares material point method (MLP-MPM) (Hu et al., 2018), whose governing equations are characterized by:

$$\rho \frac{D\mathbf{v}}{Dt} = \nabla \cdot \boldsymbol{\sigma} + \rho \mathbf{f}_{\text{ext}} \tag{1}$$

$$\rho \frac{D\rho}{Dt} = -\rho \nabla \cdot \mathbf{v}, \tag{2}$$

Table 5: Comparison to existing soft robot platforms.

| Platform | Simulation Method | Tasks | Design | Control | Differentiability | Multiphysical Materials |
|---|---|---|---|---|---|---|
| SoMoGym (Graule et al., 2022) | Rigid-link System | Mostly Manipulation | | ✓ | | |
| DiffAqua (Ma et al., 2021) | FEM | Swimmer | ✓ | ✓ | ✓ | |
| EvoGym (Bhatia et al., 2021) | 2D Mass-spring System | Locomotion Manipulation | ✓ | ✓ | | |
| SoftZoo (Ours) | MPM | Locomotion | ✓ | ✓ | ✓ | ✓ |

Table 6: The physical phenomenons that each environment covered in our multiphysical simulation.

| Environment | Elasticity | Plasticity | Fluid | Friction |
|---|---|---|---|---|
| Ground | | | | High |
| Desert | ✓ | ✓ | | |
| Wetland | ✓ | ✓ | Mixed | |
| Clay | ✓ | ✓ | | |
| Ice | | | | Low |
| Snow | ✓ | ✓ | | |
| Shallow Water | | | Shallow | |
| Ocean | | | Deep | |

where $\rho$ is the density of the material, $\mathbf{v}$ is the velocity, $\boldsymbol{\sigma}$ is the Cauchy stress of the energy, and $\mathbf{f}_{\text{ext}}$ is the external forces applied, which is the gravity in our cases. We solve these equations for an equilibrium between different materials coupled in our environments. We will not dive further into continuum mechanics and point the interested reader to Gonzalez & Stuart (2008) for more details. In terms of the implementation of the differentiable physics-based simulation, we massively use DiffTaichi (Hu et al., 2019a) as the backbone.

## E    MULTIPHYSICAL MATERIALS

For result validation and visual entertainment, we present a diverse set of environments spanned by different material setups and tasks. Here we illustrate the materials that each environment covered in Table 6. Note that even though *Desert*, *Clay*, and *Snow* share the same composition of material types, we distinguish their elastoplasticity by imposing different parameters and models (*e.g.*, friction cone). We will release the code-level implementation of all materials for reproducibility.

## F    GRADIENT CHECKPOINTING

Simulating environments like ocean, desert, etc requires a significant amount of particles. This poses a challenge in differentiation as the computation graph needs to be cached for backward pass, leading to considerably high memory usage. Accordingly, we implement gradient checkpointing that allows very large-scale simulation with gradient computation. Instead of caching simulation state at every single step, we only store data every $N$ steps. When doing backward pass at $i \times N$ steps, we perform recomputation of forward pass from $(i-1) \times N$ to $i \times N$ to reconstruct the computation graph in-between for reverse-mode automatic differentiation.

## G    OPTIMIZATION

In this section, we describe the implementation details of each optimization method.

## G.1 DIFFERENTIABLE PHYSICS

Model-based gradient provides much accurate searching direction and thus considerably more efficient optimization. However, gradient information is susceptible to local optimum and often leads to bad convergence without proper initialization. Hence, we adopt a simple yet effective approach that samples 8 random seeds, performs optimization with differentiable physics for all runs, and picks the best result. More formally we write as,

$$\theta^* = F(\mathcal{L}; \arg\min_{\theta_0} F(\mathcal{L}, \theta_0))$$

where $\theta$ are the model parameters, $\theta_0$ is the initialization of a model, $\mathcal{L}$ is a cost function, $F$ summarizes the optimization that follows the gradient update $\theta_{k+1} = \theta_k - \eta h(\nabla \mathcal{L})$, $\eta$ is the learning rate, and $h$ is a function that specifies different gradient descent variants. For the large-scale benchmark with biologically-inspired design (Table 1), we use learning rate 0.1 and training iterations 30. For all control-only and design-only optimization, we use learning rate 0.01 and training iterations 100. For co-design, we use learning rate 0.01 for both control and design with training iterations 250. We use Adam as the optimizer.

## G.2 REINFORCEMENT LEARNING

RL is only used in control optimization. We use Proximal Policy Optimization (PPO) (Schulman et al., 2017) with the following hyperparameters: number of timesteps $10^5$, buffer size 2048, batch size 32, GAE coefficient 0.95, discounting factor 0.98, number of epochs 20, entropy coefficient 0.001, learning rate 0.0001, clip range 0.2. We use the same controller parameterization as all other experiments throughout the paper.

## G.3 EVOLUTION STRATEGY

We implement a fully-ES-based method as a co-design baseline. The genome fitness function is set as the episode reward of the environment. We pose a constraint on connected component of robot body. For CPPN, we use a set of activation functions including *sigmoid*, *tanh*, *sin*, *gaussian*, *selu*, *abs*, *log*, *exp*. The inputs of CPPN include x, y, z coordinates along with distance along xy, xz, yz planes and radius from the body center. We use HyperNEAT (Stanley et al., 2009) for design optimization and CMA-ES (Hansen et al., 2003) for control optimization with initial standard deviation as 0.1. We don't use an inner-outer-loop scheme for co-design. Instead, HyperNEAT and CMA-ES share the same set of population with population size as 10. We run ES for 100 generations for the co-design baseline.

## H CONTROLLER PARAMETERIZATION

Locomotion often exhibits cyclic motion and thus control optimization can significantly benefit from considering periodic functions in controller parameterization. Specifically, we use a set of sine functions with different frequency and phase (offset) as bases. The controller is hence parameterized with a set of weights on these bases along with bias terms.

$$\bar{u}(x, t) = \sum_{ij} \alpha_{ij} \phi_{ij}(x, t) + \beta_{ij}, \quad \phi_{ij}(t) = \sin(\omega_i(x)t + \varphi_j(x))$$

where $\phi_{ij}$ are the bases, $\alpha_{ij}$ and $\beta_{ij}$ are learnable weights and biases, $x$ is controller inputs, $\omega_i(x), \varphi_j(x)$ can be either learnable (as neural network) or pre-defined. The control signal is then modulated by a tanh function multiplied by a pre-defined constant to ensure satisfaction of control bound. We use 4 different phases with frequency 20 and 80 $rad/s$ throughout the paper. While all experiments presented are not confined to using this sinewave basis controller, we empirically found it extremely efficient to generate reasonable results in comparison to other controller parameterizations like a generic neural network or trajectory optimization.

## I DESIGN SPACE REPRESENTATION

In this section, we provide more implementation details of design space representations. We first recall the notation of robot design interface. $m_p \in \mathbb{R}$ is geometry modeled by mass. $s_p \in \mathbb{R}$ is

stiffness. $\mathbf{r}_p \in \mathbb{R}^K$ is muscle placement (muscle group assignment) with $K$ as the maximal number of actuators/muscles. $\mathbf{f}_p \in \mathbb{R}^3$ is muscle direction represented in Euler angle. $\tau_m$ is the cutoff threshold of excluding low mass regions for numerical stability. Note that the subscript $p$ indicates attributes associated with a particle. $\text{MLP}_{a \to b}(\cdot; \theta)$ denotes a multi-layer perceptron with $a$ inputs, $b$ outputs, and $\theta$ as parameters (we omit hidden layers for clean notation and the output layer is linear). $m_0, s_0 \in \mathbb{R}$ are pre-defined and constant reference mass and stiffness respectively. Geometry of all methods are processed by the following formula for numerical stability,

$$m_p = \mathbb{1}_{\hat{m}_p \geq \tau_m} \hat{m}_p$$

where $\mathbb{1}$. is an indicator function. $\hat{\mathbf{x}}_p \in \mathbb{R}^3$ is a centered and standardized position $\mathbf{x}_p \in \mathbb{R}^3$. We denote the position of a base particle set $\{\hat{\mathbf{x}}_p\}$ as a point cloud representation that span the robot design workspace, which is a natural representation of continuum material in MPM. Next, we describe each design space representation individually.

### I.1 PARTICLE-BASED REPRESENTATION.

Given a base particle set $\{\hat{\mathbf{x}}_p\}$, we instantiate two trainable scalars followed by sigmoid for geometry and stiffness, and a $K$-dimensional vector followed by softmax for muscle placement with a fixed muscle direction along the canonical heading direction.

$$\hat{m}_p^{\text{PBR}} = m_0 \cdot \text{Sigmoid}(\tilde{m}_p)$$
$$s_p^{\text{PBR}} = s_0 \cdot \text{Sigmoid}(\tilde{s}_p)$$
$$\mathbf{r}_p^{\text{PBR}} = \text{Softmax}(\tilde{r}_p)$$
$$\mathbf{f}_p^{\text{PBR}} = \mathbf{f}_p^{(0)}$$

where $\{\tilde{m}_p \in \mathbb{R}, \tilde{s}_p \in \mathbb{R}, \tilde{r}_p \in \mathbb{R}^K\}$ are learnable parameters associated with $\hat{\mathbf{x}}_p$, and $\mathbf{f}_p^{(0)}$ is the fixed muscle direction along the canonical heading direction.

### I.2 VOXEL-BASED REPRESENTATION.

We voxelize the given base particle set to obtain a voxel grid and follows similar modelling technique to particle-based representation in voxel level.

$$\hat{m}_p^{\text{VBR}} = m_0 \cdot \text{Sigmoid}(\text{V2P}(\tilde{m}_v))$$
$$s_p^{\text{VBR}} = s_0 \cdot \text{Sigmoid}(\text{V2P}(\tilde{s}_v))$$
$$\mathbf{r}_p^{\text{VBR}} = \text{Softmax}(\text{V2P}(\tilde{r}_v))$$

where V2P is a mapping from voxel space to particle space (e.g., suppose we use a $2 \times 2 \times 2$ voxel grid to represent $10^3$ particles, we may associate the voxel coordinate $(0, 0, 0)$ to first $5^3$ particles), $\{\tilde{m}_v \in \mathbb{R}, \tilde{s}_v \in \mathbb{R}, \tilde{r}_v \in \mathbb{R}^K\}$ are learnable parameters, and the muscle direction is of the same definition as particle-based representation $\mathbf{f}_p^{\text{VBR}} = \mathbf{f}_p^{\text{PBR}}$. In comparison to particle-based representations, in practice the voxel-based representations have much fewer parameters that need to be learned.

### I.3 IMPLICIT FUNCTION

We extend the idea of OccupancyNet (Mescheder et al., 2019) to predicting robot geometry, stiffness, and muscle placement. It is modeled by a multi-layer perceptron (MLP) with 2 layers and 32 dimensions for each. We use *tanh* activation. The MLP takes in x, y, z coordinates, distance along xy, xz, yz planes and radius from the body center. The network outputs occupancy as geometry using sigmoid, stiffness multiplier using sigmoid, and a $K$-dimensional vector using softmax for muscle placement. We describe in formula as,

$$\hat{m}_p^{\text{IF}} = m_0 \cdot \text{Sigmoid}(\text{MLP}_{3 \to 1}(\hat{\mathbf{x}}_p; \theta_m))$$
$$s_p^{\text{IF}} = s_0 \cdot \text{Sigmoid}(\text{MLP}_{3 \to 1}(\hat{\mathbf{x}}_p; \theta_s))$$
$$\mathbf{r}_p^{\text{IF}} = \text{Softmax}(\text{MLP}_{3 \to K}(\hat{\mathbf{x}}_p; \theta_\mathbf{r}))$$

where $\{\theta_m, \theta_s, \theta_\mathbf{r}\}$ are learnable parameters formulated as neural networks, and the muscle direction is of the same definition as particle-based representation $\mathbf{f}_p^{\text{IF}} = \mathbf{f}_p^{\text{PBR}}$. This representation has inherent spatial continuity due to the use of smooth functions MLP with spatial coordinate $\hat{\mathbf{x}}_p$ as inputs.

## I.4   DIFF-CPPN

*Diff-CPPN* is a differentiable version of Compositional Pattern Producing Networks (CPPN) (Stanley, 2007), following similar concept in (Fernando et al., 2016). CPPN is a graphical model $\mathcal{G} = \{\mathcal{N}, \mathcal{E}\}$ composed of a set of activation functions $\Sigma$ with interesting geometric properties (e.g., sine, tanh) that takes in particle or voxel coordinates and output occupancy or other properties. Each node $n_i \in \mathcal{N}$ has a set of input edges $e_i \in \mathcal{E}$ that can be changed by evolution, and an activation function $\sigma_i \in \sum$. The input-output relationship of a layer can be then written as,

$$n_i^{\text{out}} = \sigma_i(\sum_{e_j \in \mathcal{E}} w_j n_j^{\text{in}})$$

It is originally designed to be optimized with varying graph topologies. We use a meta graph to allow gradient flow and mimic the augmenting topolgies process in NEAT yet in a differentiable manner, i.e., the varying topology $w_j \in \{0, 1\}$ is softened to $w_j \in \mathbb{R}$. Remark the similarity of the above construction with a layer in regular MLP except for varying activation function across neurons. We then can define,

$$\hat{m}_p^{\text{Diff-CPPN}} = m_0 \cdot \text{Sigmoid}(\text{MLP}_{3\to1}^{\text{CPPN}}(g(\hat{\mathbf{x}}_p); \theta_m))$$
$$s_p^{\text{Diff-CPPN}} = s_0 \cdot \text{Sigmoid}(\text{MLP}_{3\to1}^{\text{CPPN}}(g(\hat{\mathbf{x}}_p); \theta_s))$$
$$\mathbf{r}_p^{\text{Diff-CPPN}} = \text{Softmax}(\text{MLP}_{3\to K}^{\text{CPPN}}(g(\hat{\mathbf{x}}_p); \theta_{\mathbf{r}}))$$

where each layer of $\text{MLP}^{\text{CPPN}}$ follows the aforementioned input-output relationship, $\{\theta_m, \theta_s, \theta_{\mathbf{r}}\}$ are learnable parameters, and $g(\cdot)$ is a function of expanding position to additional spatial coordinates such as distance to the center (which is normally used in CPPN). We use a fixed muscle direction.

The model takes in x, y, z coordinates, distance along xy, xz, yz planes and radius from the body center, and outputs occupancy as geometry using sigmoid, stiffness multiplier using sigmoid, and a $K$-dimensional vector using softmax for muscle placement. We use *sin* and *sigmoid* activation functions with 3 hidden layers and 20 graph nodes in each layer.

## I.5   SDF-LERP

Given a base particle set $\{\hat{\mathbf{x}}_p\}$, we compute the linear interpolation among a set of pre-defined design primitives $\{\Psi_i\}_{i=1}^N$, where $N$ is number of design primitives. The shape of the robot design $m_p$ is then determined by a set of coefficients weighting the SDFs from design primitives. In other words, the trainable parameters for robot geometry only construct a $N$-dimensional vector. We then compute weighted sum of the SDF bases and extract robot body with the final SDF smaller or equal to zero. We use a low-temperature sigmoid in implementation to keep gradient flow. For stiffness $s_p$, we can directly perform linear interpolation. For muscle group membership $\mathbf{r}_p$, we use linear interpolation upon the one-hot vectors from design primitives and effectively realize a soft muscle group assignment. For muscle direction $\mathbf{f}_p$, we adopt interpolation designed for rotation matrices (Brégier, 2021). Formally,

$$\hat{m}_p^{\text{SDF-Lerp}} = m_0 \cdot \frac{1}{1 + e^{-T \sum_{i=1}^N \hat{\alpha}_i \Psi_i^{\text{SDF}}(\hat{\mathbf{x}}_p)}}, \quad \hat{\alpha}_i = \frac{\alpha_i}{\sum_j \alpha_j}$$

$$s_p^{\text{SDF-Lerp}} = \sum_{i=1}^N \hat{\beta}_i \Psi_i^s(\hat{\mathbf{x}}_p), \quad \hat{\beta}_i = \frac{\beta_i}{\sum_j \beta_j}$$

$$\mathbf{r}_p^{\text{SDF-Lerp}} = \sum_{i=1}^N \hat{\gamma}_i \Psi_i^{\mathbf{r}}(\hat{\mathbf{x}}_p), \quad \hat{\gamma}_i = \frac{\gamma_i}{\sum_j \gamma_j}$$

$$\mathbf{f}_p^{\text{SDF-Lerp}} = \text{M2E}(\underset{\mathbf{R} \in \text{SO}(3)}{\arg\min} ||\mathbf{R} - \sum_{i=1}^N \hat{\kappa}_i \text{E2M}(\Psi_i^{\mathbf{f}}(\hat{\mathbf{x}}_p))||_F^2), \quad \hat{\kappa}_i = \frac{\kappa_i}{\sum_j \kappa_j}$$

where $\{\alpha \in \mathbb{R}^N, \beta \in \mathbb{R}^N, \gamma \in \mathbb{R}^N, \kappa \in \mathbb{R}^N\}$ are learnable coefficients of the interpolation, $\Psi_i^{\text{SDF}}, \Psi_i^s, \Psi_i^{\mathbf{r}}, \Psi_i^{\mathbf{f}}$ are the SDF, stiffness, muscle placement, and muscle direction of a design primitive respectively, $T$ is temperature (where we set to $-1000$ to mimic SDF $\leq 0$), E2M is conversion from an Euler angle to a rotation matrix, M2E is conversion from a rotation matrix to an Euler angle, $|| \cdot ||_F$ is Frobenius norm. The computation of muscle direction $\mathbf{f}_p^{\text{SDF-Lerp}}$ follows rotation

Table 7: Full RL results of Table 1. Each entry is the mean and standard deviation of 5 random seeds. The higher the better.

| Task | Animal | Ground | Ice | Wetland | Environment Clay | Desert | Snow | Ocean |
|------|--------|--------|-----|---------|------|--------|------|-------|
| Movement Speed | Baby Seal | $0.154 \pm 0.096$ | $0.010 \pm 0.010$ | $0.020 \pm 0.004$ | $0.005 \pm 0.002$ | $0.034 \pm 0.010$ | $0.016 \pm 0.011$ | $0.029 \pm 0.003$ |
| | Caterpillar | $0.032 \pm 0.020$ | $0.006 \pm 0.005$ | $0.016 \pm 0.005$ | $0.015 \pm 0.004$ | $0.032 \pm 0.004$ | $0.017 \pm 0.003$ | $0.181 \pm 0.014$ |
| | Fish | $0.033 \pm 0.012$ | $0.011 \pm 0.009$ | $0.013 \pm 0.003$ | $0.014 \pm 0.003$ | $0.022 \pm 0.005$ | $0.042 \pm 0.017$ | $0.151 \pm 0.011$ |
| | Panda | $0.019 \pm 0.008$ | $0.006 \pm 0.005$ | $0.008 \pm 0.002$ | $0.005 \pm 0.001$ | $0.009 \pm 0.002$ | $0.004 \pm 0.001$ | $0.007 \pm 0.003$ |
| Turning | Baby Seal | $0.077 \pm 0.020$ | $0.024 \pm 0.009$ | $0.008 \pm 0.003$ | $0.011 \pm 0.001$ | $0.026 \pm 0.011$ | $0.028 \pm 0.009$ | $0.020 \pm 0.026$ |
| | Caterpillar | $0.021 \pm 0.007$ | $0.009 \pm 0.005$ | $0.006 \pm 0.002$ | $0.005 \pm 0.003$ | $0.015 \pm 0.008$ | $0.006 \pm 0.005$ | $0.358 \pm 0.024$ |
| | Fish | $0.047 \pm 0.010$ | $0.012 \pm 0.009$ | $0.010 \pm 0.006$ | $0.007 \pm 0.002$ | $0.019 \pm 0.008$ | $0.029 \pm 0.019$ | $0.013 \pm 0.005$ |
| | Panda | $0.014 \pm 0.007$ | $0.003 \pm 0.003$ | $0.004 \pm 0.000$ | $0.001 \pm 0.000$ | $0.002 \pm 0.002$ | $0.003 \pm 0.002$ | $0.031 \pm 0.001$ |
| Velocity Tracking | Baby Seal | $0.410 \pm 0.236$ | $0.194 \pm 0.079$ | $0.222 \pm 0.026$ | $0.205 \pm 0.024$ | $0.265 \pm 0.024$ | $0.198 \pm 0.041$ | $0.068 \pm 0.051$ |
| | Caterpillar | $0.368 \pm 0.081$ | $0.156 \pm 0.358$ | $0.192 \pm 0.034$ | $0.058 \pm 0.028$ | $0.376 \pm 0.047$ | $0.133 \pm 0.221$ | $0.714 \pm 0.072$ |
| | Fish | $0.256 \pm 0.113$ | $0.319 \pm 0.123$ | $0.236 \pm 0.026$ | $0.303 \pm 0.070$ | $0.311 \pm 0.157$ | $0.307 \pm 0.127$ | $0.574 \pm 0.141$ |
| | Panda | $0.424 \pm 0.049$ | $0.383 \pm 0.139$ | $0.534 \pm 0.082$ | $0.195 \pm 0.063$ | $0.359 \pm 0.034$ | $0.288 \pm 0.073$ | $0.220 \pm 0.268$ |
| Waypoint Following | Baby Seal | $-0.014 \pm 0.008$ | $-0.027 \pm 0.002$ | $-0.027 \pm 0.001$ | $-0.026 \pm 0.001$ | $-0.025 \pm 0.002$ | $-0.026 \pm 0.001$ | $-0.026 \pm 0.001$ |
| | Caterpillar | $-0.016 \pm 0.004$ | $-0.028 \pm 0.003$ | $-0.028 \pm 0.000$ | $-0.027 \pm 0.000$ | $-0.027 \pm 0.001$ | $-0.026 \pm 0.001$ | $-0.019 \pm 0.007$ |
| | Fish | $-0.016 \pm 0.005$ | $-0.026 \pm 0.002$ | $-0.029 \pm 0.003$ | $-0.027 \pm 0.003$ | $-0.024 \pm 0.001$ | $-0.024 \pm 0.001$ | $-0.023 \pm 0.000$ |
| | Panda | $-0.014 \pm 0.004$ | $-0.025 \pm 0.004$ | $-0.027 \pm 0.001$ | $-0.028 \pm 0.000$ | $-0.024 \pm 0.001$ | $-0.025 \pm 0.001$ | $-0.026 \pm 0.000$ |

matrix interpolation using special Procrustes method, detailed in (Brégier, 2021). In comparison to the above methods, *SDF-Lerp* leverages prior knowledge of design primitives $\Psi_i$ that can provide more structure to the optimization from design space representation.

## I.6 Wasserstein Barycenter

This method also uses design primitives $\{\Psi_i\}_{i=1}^N$. First, it adopts the same approach for stiffness and muscle placement as *SDF-Lerp* , $s_p^{\text{Wass}} = s_p^{\text{SDF-Lerp}}, \mathbf{r}_p^{\text{Wass}} = \mathbf{r}_p^{\text{SDF-Lerp}}$. We use a fixed muscle direction along the canonical heading direction. The major difference is the way to represent robot geometry. Following (Ma et al., 2021), we define a probability simplex (i.e., a set of coefficients with length as the number of design primitives) that serves as a weighting in the sense of Wasserstein distance among different shapes. It can be written as,

$$\hat{m}_p^{\text{Wass}} = m_0 P_\alpha(\hat{\mathbf{x}}_p)$$

where $P_\alpha$ is the Wasserstein barycenter with coefficients $\alpha$ (intuitively, with some abuse of notation, we can view it as probability density function that determines the particle existence), defined as,

$$P_\alpha = \sum_P \sum_i \alpha_i \mathcal{W}_2^2(P, \Psi_i)$$

$$\mathcal{W}_2(P, Q) = \left[ \inf_{\pi \in \prod(P,Q)} \iint (\mathbf{x}_p - \mathbf{x}_q)^2 d\pi(\mathbf{x}_p, \mathbf{x}_q) \right]^{\frac{1}{2}}$$

where $\mathcal{W}_2$ is 2-Wasserstein distance and $\Psi_i$ is shape primitive. This representation better preserves the volume from the shape of design primitives. We refer the reader to the original paper for more details.

## J Full RL Results

In Table 1, we show large-scale benchmark of biologically-inspired designs in SoftZoo. The comparison includes optimization using differentiable physics and RL. We use 5 random seeds for RL results yet only report average performance due to space limit. In Table 7, we report the full RL results with both mean and standard deviation among all seeds.

## K Visualization of Biologically-inspired Design

In this section, we demonstrate the biologically-inspired designs used in this paper. In Figure 7, we show the four animals used in the main paper, including *Baby Seal*, *Caterpillar*, *Fish*, and *Panda*. In Figure 8, we show the set of design primitives used in *SDF-Lerp* and *Wasserstein Barycenter*.

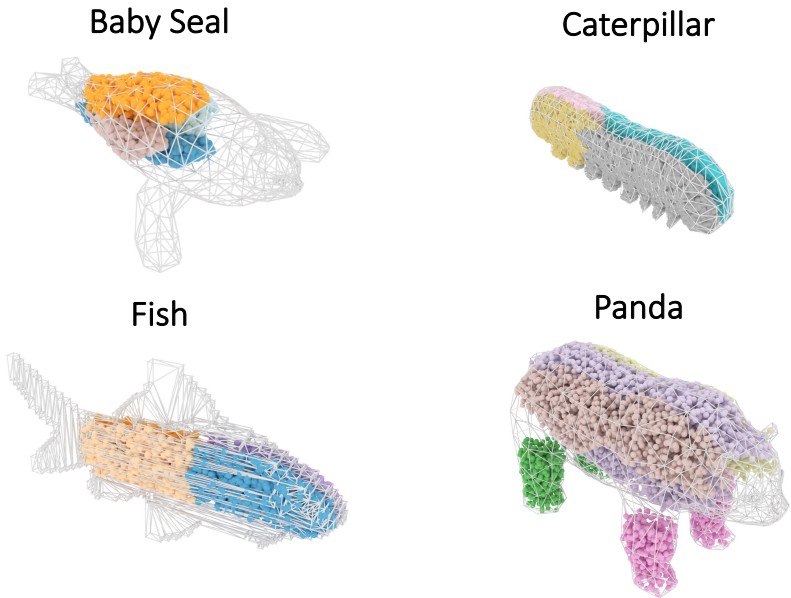

Figure 7: Visualization of animals for biologically-inspired design.

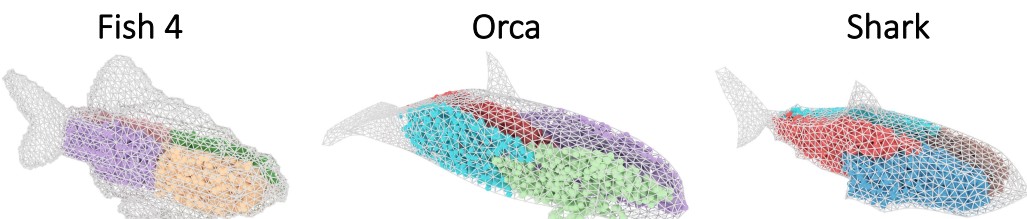

Figure 8: Visualization of fish-like design primitives.

