# OpenReview forum: "SoftZoo: A Soft Robot Co-design Benchmark For Locomotion In Diverse Environments"
_ICLR.cc/2023/Conference — ICLR 2023 poster_

### Official Review · Reviewer_wyCA · 2022-10-23

**Confidence:** 3
**Correctness:** 4
**Technical Novelty And Significance:** 3
**Empirical Novelty And Significance:** 3
**Recommendation:** 8

**Clarity, Quality, Novelty And Reproducibility:**

The paper clearly described the proposed platform and the various experiments. Of course, I expect the code to be released so that people can build on this research.

**Strength And Weaknesses:**

Pro:

1. Introduction of a platform for design/control co-optimization for soft robots.

2. Large-scale experiments with insight that can guide future research direction on this topic.

Con:

It is unclear whether the simulated platform is accurate enough such that the design and control coming out of the optimization can actually be used for real robots.

**Summary Of The Paper:**

This paper presents a platform for soft robot codesign research. This platform provides a wide range of environments and tasks. Large-scale experiments are performed on design and control optimization.

**Summary Of The Review:**

This paper provides a useful platform to conduct research in soft robot design and control. Experiments are shown to showcase the various features of the platform and provide insights and suggestions for future research direction. Although it is unclear whether the proposed platform can be useful for real robot design, I believe this is still a nice contribution to the learning design/control community.

---

> ### Author Response · Authors · 2022-11-16
> **Response to Reviewer wyCA - Part 1**
>
> We thank reviewer wyCA for acknowledging the contribution and impact of our work. We address the remaining suggestions as the below.
>
> >**Usefulness for real robots.**
>
> We thank the reviewer for raising this point. Simulation platforms have served as a driving force for the development of algorithms and learning in various robotic applications [10,11]. SoftZoo provides a platform to, from a more algorithmic perspective such as representation learning and optimization, extensively study how to push forward research of soft robot co-design. Our current observations and more future findings from the community using this platform may provide useful insight into building physical robots. These insights could include:
> - Algorithm developments, such as design representations or optimization techniques shown in our work as opposed to the actual instance of optimized robot design and control.
> - Understanding of the interplay between environment, morphology, and behavior from a computational perspective.
>
> Sim-to-real for soft robotics is an extremely challenging problem; presently, entire manuscripts are focused on solving control or system identification problems for very simple and singular structures [5,9]. The ability to transfer from simulation to reality robustly for ANY soft robot would be a monumental achievement for the field and not addressable within this manuscript.  Some notable challenges of sim-to-real transfer include,
> - System identification to determine physical parameters of the simulation.
> - The accuracy of material models, multiphysical coupling, and contact handling.
> - Feasibility of robot fabrication.
>
> We highlight several promising directions that can potentially remedy the above challenges.
> - Differentiable physics has recently become a prominent direction for system identification of deformable objects [2,3], rigid-bodied robots [4], and soft robots [1,5]. The idea is to use gradients in differentiable simulation to estimate the physical parameters of physical systems. With this type of solution, we can “calibrate” SoftZoo such that it mostly resembles real fabricated soft robots.
> - Domain randomization has been a workhorse of sim-to-real in rigid robotics [6,7]. It injects randomness in simulation during training to achieve robustness of learned models for handling sim-to-real gaps. While there is limited research done for soft bodies [8], this concept can be useful on its own or complementary to other sim-to-real methods.
> - One limitation of our simulated robots is that we do not consider the fabrication constraints of potential physical counterparts; for instance, in order to fabricate an anisotropic pneumatic silicone chamber that contracts/extends or bends (such as a PneuNet structure), one must ensure the region is large enough to have sufficient wall-thickness so that it is structural, that it has a hollow (and light) core, and, typically to program the anisotropy, it must have a ribbed structure. Different fabrication constraints are appropriate at different scales, constitutive material choices, and fabrication methods. While it would be very useful to automatically enforce feasibility of fabrication or reject designs in which fabrication constraints are violated, that would be the start of a lengthy experimental journey that is most appropriate in a paper on the targeted fabrication method itself. We could add a discussion about this to the paper, if the reviewers think it would be useful.
>
> While SoftZoo provides insights into a more algorithmic development of soft robot co-design, there is much more future work left to be done to bridge the gap between simulation and physical robots.

---

> > ### Author Response · Authors · 2022-11-16
> > **Response to Reviewer wyCA - Part 2**
> >
> >
> > **Reference**
> >
> > [1] Hu, Y., Liu, J., Spielberg, A., Tenenbaum, J.B., Freeman, W.T., Wu, J., Rus, D. and Matusik, W., 2019, May. Chainqueen: A real-time differentiable physical simulator for soft robotics. In 2019 International conference on robotics and automation (ICRA) (pp. 6265-6271). IEEE.
> >
> > [2] Sundaresan, P., Antonova, R. and Bohg, J., 2022. DiffCloud: Real-to-Sim from Point Clouds with Differentiable Simulation and Rendering of Deformable Objects. arXiv preprint arXiv:2204.03139.
> >
> > [3] Cleac'h, S.L., Yu, H.X., Guo, M., Howell, T.A., Gao, R., Wu, J., Manchester, Z. and Schwager, M., 2022. Differentiable Physics Simulation of Dynamics-Augmented Neural Objects. arXiv preprint arXiv:2210.09420.
> >
> > [4] Ma, P., Du, T., Tenenbaum, J.B., Matusik, W. and Gan, C., 2022. RISP: Rendering-Invariant State Predictor with Differentiable Simulation and Rendering for Cross-Domain Parameter Estimation. arXiv preprint arXiv:2205.05678.
> >
> > [5] Du, T., Hughes, J., Wah, S., Matusik, W. and Rus, D., 2021. Underwater soft robot modeling and control with differentiable simulation. IEEE Robotics and Automation Letters, 6(3), pp.4994-5001.
> >
> > [6] Tobin, J., Fong, R., Ray, A., Schneider, J., Zaremba, W. and Abbeel, P., 2017, September. Domain randomization for transferring deep neural networks from simulation to the real world. In 2017 IEEE/RSJ international conference on intelligent robots and systems (IROS) (pp. 23-30). IEEE.
> >
> > [7] Peng, X.B., Andrychowicz, M., Zaremba, W. and Abbeel, P., 2018, May. Sim-to-real transfer of robotic control with dynamics randomization. In 2018 IEEE international conference on robotics and automation (ICRA) (pp. 3803-3810). IEEE.
> >
> > [8] Matas, J., James, S. and Davison, A.J., 2018, October. Sim-to-real reinforcement learning for deformable object manipulation. In Conference on Robot Learning (pp. 734-743). PMLR.
> >
> > [9] Dubied, M., Michelis, M.Y., Spielberg, A. and Katzschmann, R.K., 2022. Sim-to-Real for Soft Robots Using Differentiable FEM: Recipes for Meshing, Damping, and Actuation. IEEE Robotics and Automation Letters, 7(2), pp.5015-5022.
> >
> > [10] Dosovitskiy, A., Ros, G., Codevilla, F., Lopez, A. and Koltun, V., 2017, October. CARLA: An open urban driving simulator. In Conference on robot learning (pp. 1-16). PMLR.
> >
> > [11] Shah, S., Dey, D., Lovett, C. and Kapoor, A., 2018. Airsim: High-fidelity visual and physical simulation for autonomous vehicles. In Field and service robotics (pp. 621-635). Springer, Cham.
> >
> > Thanks, authors

---

### Official Review · Reviewer_yNAY · 2022-10-23

**Confidence:** 4
**Correctness:** 4
**Technical Novelty And Significance:** 2
**Empirical Novelty And Significance:** 2
**Recommendation:** 6

**Clarity, Quality, Novelty And Reproducibility:**

The quality of the paper is good with many results and baselines.
The clarity of the paper is a bit on the low side, and needs rewriting a large portion to improve its readability.
The originality of the paper is good as a system paper.


**Strength And Weaknesses:**

Strengths of the paper:
1) A thorough paper with plenty of results (i.e. large scale benchmarks for various robot morphologies and environments.
2) Many baselines are compared for the design space choices, which makes the conclusion more convincing.
3) The accompanying video provides

Weaknesses of the paper:
1) The paper is overall not clearly written with many missing details and leaves the readers to guess. For example, section 3.2 discussed various baselines used to perform gradient based design optimization. However, even combining the appendix it is still difficult to understand how the design optimization is actually performed. Another example is the colored muscle groups. The word “muscle group” has appeared twice in the paper without a proper definition. I assume this has something to do with the design optimization.
2) The website is not well constructed with many illustrations missing.


**Summary Of The Paper:**

In this paper, the authors introduced a framework (SoftZoo) that can co-design the morphology and controller for soft robots in diverse environments. SoftZoo explores a differentiable material point method (MPM) to simulate soft bodies and their interaction with different terrain types, and the soft robots are actuated through anisotropic elastic energy.  The authors constructed a set of biologically inspired soft robots and benchmarked their performances in various environments, and discussed the importance of design space representations and the ambiguity between design and control optimization.

**Summary Of The Review:**

Overall the paper provides interesting (and large scale) results with plenty of baselines; The paper is not well written and needs improvement in its clarity.

---

> ### Author Response · Authors · 2022-11-16
> **Response to Reviewer yNAY**
>
> We appreciate reviewer yNAY for recognizing the quality and thoroughness of our work, and providing constructive feedback. We address the remaining concern as below.
>
> >**The paper is not clearly written with many missing details.**
>
> We thank the reviewer for bringing up this issue. We improve the clarity with the following revision in the updated manuscript (highlighted in red).
> - In the main paper Section 2.2 and 2.3, we add more description along with the use of additional notations that are associated with the simulation and implementation details.
> - We add a new section in appendix (Appendix A) to give a list of notation used in the entire manuscript.
> - In Appendix B, we detailedly write down the formulation of metrics used in all tasks including movement speed, turning, velocity tracking, and waypoint following.
> - In Appendix G.1, we expand the description of optimization using differentiable physics.
> - In Appendix H, we write a more detailed formulation of the controller parameterization.
> - In Appendix I, we greatly revise the text to include more detailed descriptions of design optimization baselines based on the robot design interface of SoftZoo and introduce the definition of muscle group.
>
> Please let us know if there are further areas for which more detail would improve the clarity of the manuscript and we will be happy to make those improvements.
>
> >**Many illustrations on the website are missing.**
>
> Thanks for reporting this issue. We manually stress-tested our website and found that while it worked properly most of the time, some illustrations failed to load at some trials. This is because the gif files are high resolution and thus quite large. We have improved the website by avoiding loading many animations at once for a better viewing experience.
> - https://sites.google.com/view/softzoo-iclr-2023
>
> Let us know if this resolves the issue.
>
> Thanks, authors

---

### Official Review · Reviewer_zLWw · 2022-10-24

**Confidence:** 2
**Correctness:** 4
**Technical Novelty And Significance:** 3
**Empirical Novelty And Significance:** 3
**Recommendation:** 8

**Clarity, Quality, Novelty And Reproducibility:**

Clarity: The paper is clearly written, and experimental details are described thoroughly.

Quality: The experiments are thorough, although I would hope to see additional random seeds from RL experiments (although I understand that this is computationally very heavy).

Novelty: As mentioned in strengths, existing frameworks for soft robot co-design either are not differentiable or are relatively narrow in task selection. Additionally, this paper presents novel analysis in its empirical studies.

Nitpick: Typo at the end of section 4: “robitics”

**Strength And Weaknesses:**

Strengths:
- The paper is well motivated, and the proposed platform seems like it will help to drive research in the direction of developing improved co-design algorithms. It supports a variety of design parameterizations and diverse terrain types.
- Compared to prior platforms, SoftZoo is differentiable (unlike EvoGym) and supports more tasks (unlike DiffAqua) but only supports locomotion tasks (a weakness).
- The experimental analysis is very insightful. I think that they highlight several of the most critical issues currently in the co-optimization of design and control.
- The presentation of the paper is clear and it is well written. The visualizations of the demos are really nice!

Weaknesses:
- As mentioned earlier, the tasks are all locomotion tasks. Therefore their complexity is somewhat limited, especially because the different terrains may not require as much specialization in design compared to, say, manipulation tasks.


**Summary Of The Paper:**

This paper presents a framework called SoftZoo for co-learning design and control of soft robots in a variety of locomotion environments. It includes the design parameterizations for designs and control policies, task definitions, environments created in the TaiChi differentiable simulator, and prior co-design algorithms. The paper also introduces experiments to analyze the problem of soft robot co-design from multiple perspectives: how biologically inspired morphologies perform in different environments, choice of design spaces, muscle stiffness, and the use of differentiable simulation. Finally they demonstrate co-optimization for a robotic swimmer.


**Summary Of The Review:**

In summary, this paper presents a benchmark built on a differentiable simulator for the co-optimization of design and control of soft robots. It allows testing locomotion tasks in a variety of terrains. The paper is well-executed and provides valuable analysis, and I think this benchmark will be a nice contribution to the field.

---

> ### Author Response · Authors · 2022-11-16
> **Response to Reviewer zLWw - Part 1**
>
> We thank reviewer zLWw for positive comments on all aspects including clarity, quality, and novelty. We address the remaining suggestions as below.
>
> >**The platform supports locomotion tasks in diverse environments, but doesn’t include other types of tasks like manipulation.**
>
> We thank the reviewer for bringing up this concern. We acknowledge that locomotion is only one subset of soft robotic tasks of interest. Note, however, that our focus on locomotion does not preclude SoftZoo as being used for other soft robotics problems, such as manipulation. In light of conducting further study on other types of tasks, we demonstrate an example of a manipulation task: the goal of a manipulator is to throw a snowball as far as possible as shown in the video,
> - Google drive video: https://drive.google.com/file/d/1EFUStBCPjRhgFFlwMjO-eNVUv3RgtXOR/view?usp=share_link
> - Video on website: https://sites.google.com/view/softzoo-iclr-2023/simple-extension-to-manipulation?authuser=0
>
> We use a 3D model of a human arm with a hand and cropped fingers as the robot shape. We manually label four muscle fibers on the arm along the vertical direction. We perform trajectory optimization to maximize the traverse distance of the snowball using differentiable physics. This simple yet interesting illustration shows the potential of extending SoftZoo to other tasks like manipulation.
>
> Locomotion tasks, especially those with coupled physics and varied objectives, go beyond previous studies. Locomotion is widely considered in soft robot co-design literature [1,2,3] yet is mostly limited to simple environments like a flat ground. In this work, we aim to more extensively study this topic in the presence of diverse environments e.g., unique motions arising from morphology and environment as shown in Figure 2.
>
> On the other hand, manipulation raises its own unique challenges that are deserving of their own dedicated manuscript.  For example, manipulation relies on geometrically complex, fine contact.  Since contact in MPM is resolved at the grid level, an adaptive grid or a grid-free contact resolution method would be needed to capture the rich complexity of manipulation tasks.  Further, manipulation tasks are particularly constrained by the load that can be borne by the manipulator.  In MPM, a manipulator that is not load bearing may fracture, leading to unstable gradients.  Optimization methods that can reason about or work around these simulation difficulties would be needed. Last but not least, soft robotic manipulation may be of great interest for applications with objects that are fragile, vulnerable to abrasion, etc, leading to an entirely different aspect of developing algorithms and representations.
>
> >**Typos.**
>
> Thank you for pointing these out. We have fixed these typos in the revised version.
>
> **Reference**
>
> [1] Cheney, N., MacCurdy, R., Clune, J. and Lipson, H., 2014. Unshackling evolution: evolving soft robots with multiple materials and a powerful generative encoding. ACM SIGEVOlution, 7(1), pp.11-23.
>
> [2] Lipson, H., 2014. Challenges and opportunities for design, simulation, and fabrication of soft robots. Soft Robotics, 1(1), pp.21-27.
>
> [3] Cheney, N., Clune, J. and Lipson, H., 2014, July. Evolved electrophysiological soft robots. In ALIFE 14: The Fourteenth International Conference on the Synthesis and Simulation of Living Systems (pp. 222-229). MIT Press.

---

> > ### Author Response · Authors · 2022-11-16
> > **Response to Reviewer zLWw - Part 2**
> >
> > >**Additional random seeds for RL experiments.**
> >
> > We appreciate the thoughtful suggestion. We agree that RL experiments with more random seeds provide more solid results for this paper. The following table shows the average performance along with standard deviation of 5 random seeds.
> >
> > *Movement Speed*
> > |Animal|Ground|Ice|Wetland|Clay|Desert|Snow|Ocean|
> > |:---|:---:|:---:|:---:|:---:|:---:|:---:|:---:|
> > |Baby Seal|0.154 +- 0.096|0.010 +- 0.010|0.020 +- 0.004|0.005 +- 0.002|0.034 +- 0.010|0.016 +- 0.011|0.029 +- 0.003
> > |Caterpillar|0.032 +- 0.020|0.006 +- 0.005|0.016 +- 0.005|0.015 +- 0.004|0.032 +- 0.004|0.017 +- 0.003|0.181 +- 0.014
> > |Fish|0.033 +- 0.012|0.011 +- 0.009|0.013 +- 0.003|0.014 +- 0.003|0.022 +- 0.005|0.042 +- 0.017|0.151 +- 0.011
> > |Panda|0.019 +- 0.008|0.006 +- 0.005|0.008 +- 0.002|0.005 +- 0.001|0.009 +- 0.002|0.004 +- 0.001|0.007 +- 0.003
> >
> > *Turning*
> > |Animal|Ground|Ice|Wetland|Clay|Desert|Snow|Ocean|
> > |:---|:---:|:---:|:---:|:---:|:---:|:---:|:---:|
> > |Baby Seal|0.077 +- 0.020|0.024 +- 0.009|0.008 +- 0.003|0.011 +- 0.001|0.026 +- 0.011|0.028 +- 0.009|0.020 +- 0.026
> > |Caterpillar|0.021 +- 0.007|0.009 +- 0.005|0.006 +- 0.002|0.005 +- 0.003|0.015 +- 0.008|0.006 +- 0.005|0.358 +- 0.024
> > |Fish|0.047 +- 0.010|0.012 +- 0.009|0.010 +- 0.006|0.007 +- 0.002|0.019 +- 0.008|0.029 +- 0.019|0.013 +- 0.005
> > |Panda|0.014 +- 0.007|0.003 +- 0.003|0.004 +- 0.000|0.001 +- 0.000|0.002 +- 0.002|0.003 +- 0.002|0.031 +- 0.001
> >
> > *Velocity Tracking*
> > |Animal|Ground|Ice|Wetland|Clay|Desert|Snow|Ocean|
> > |:---|:---:|:---:|:---:|:---:|:---:|:---:|:---:|
> > |Baby Seal|0.410 +- 0.236|0.194 +- 0.079|0.222 +- 0.026|0.205 +- 0.024|0.265 +- 0.024|0.198 +- 0.041|0.068 +- 0.051
> > |Caterpillar|0.368 +- 0.081|0.156 +- 0.358|0.192 +- 0.034|0.058 +- 0.028|0.376 +- 0.047|0.133 +- 0.221|0.714 +- 0.072
> > |Fish|0.256 +- 0.113|0.319 +- 0.123|0.236 +- 0.026|0.303 +- 0.070|0.311 +- 0.157|0.307 +- 0.127|0.574 +- 0.141
> > |Panda|0.424 +- 0.049|0.383 +- 0.139|0.534 +- 0.082|0.195 +- 0.063|0.359 +- 0.034|0.288 +- 0.073|0.220 +- 0.268
> >
> > *Waypoint Following*
> > |Animal|Ground|Ice|Wetland|Clay|Desert|Snow|Ocean|
> > |:---|:---:|:---:|:---:|:---:|:---:|:---:|:---:|
> > |Baby Seal|-0.014 +- 0.008|-0.027 +- 0.002|-0.027 +- 0.001|-0.026 +- 0.001|-0.025 +- 0.002|-0.026 +- 0.001|-0.026 +- 0.001
> > |Caterpillar|-0.016 +- 0.004|-0.028 +- 0.003|-0.028 +- 0.000|-0.027 +- 0.000|-0.027 +- 0.001|-0.026 +- 0.001|-0.019 +- 0.007
> > |Fish|-0.016 +- 0.005|-0.026 +- 0.002|-0.029 +- 0.003|-0.027 +- 0.003|-0.024 +- 0.001|-0.024 +- 0.001|-0.023 +- 0.000
> > |Panda|-0.014 +- 0.004|-0.025 +- 0.004|-0.027 +- 0.001|-0.028 +- 0.000|-0.024 +- 0.001|-0.025 +- 0.001|-0.026 +- 0.000
> >
> > In the revised manuscript, we updated the large-scale benchmark in Table 1 with the new results and included the full results to Appendix J.
> >
> > Thanks, authors

---

### Author Response · Authors · 2022-11-16
**General Response**

We thank all reviewers for their thoughtful and constructive feedback. We are encouraged to hear the reviewers,
- acknowledge the proposed platform as well-motivated (Reviewer zLWw) and useful (Reviewer wyCA) to drive research in the direction of developing co-design algorithms,
- and that the experimental analysis is thorough (Reviewers zLWw, yNAY, wyCA) and insightful (Reviewers zLWw, wyCA) to highlight critical issues and guide future research directions,
- and that our work is of good quality (Reviewer yNAY) with a nice contribution to the community (Reviewers zLWw, wyCA).

In response to feedback, we provide individual responses below to address the remaining concerns from each reviewer and an updated manuscript with changes highlighted in red to improve clarity of missing details. Briefly, we summarize the added experiments and revision to the paper,
- RL experiments with additional random seeds (Reviewer zLWw).
- An example of a manipulation task to demonstrate possible extension of SoftZoo along with discussion on potential future direction (Reviewer zLWw).
- Revision on paper with additional details to improve clarity (Reviewer yNAY).
- Updated website that is better constructed for improved viewing experience (Reviewer yNAY).
- Discussion on potential extension and challenges to physical robots (Reviewer wyCA).

For more details, please check individual responses. We thank all reviewers’ for their time and efforts! We hope our responses have persuasively addressed all remaining concerns. Please don’t hesitate to let us know of any additional comments or feedback on improvement.

Thanks, authors

---

### Decision · Program_Chairs · 2023-01-20

**Decision:**

Accept: poster

**Justification For Why Not Higher Score:**

The paper while it is interesting to its community, may not have larger interests to warrant a spotlight or oral presentation. This is a judgement call and I don't have further reason not to give it a spotlight presentation.

**Justification For Why Not Lower Score:**

The paper introduces new tooling and and provides experiments to highlight the important areas for research. While it may not be the most exciting paper, it achieves its goals and can provide value to others.

**Metareview: Summary, Strengths And Weaknesses:**

This paper introduces a differentiable simulator for co-designing the morphology and controller for soft robots in diverse environments. In addition to introducing the new benchmark, the paper also provides experiments with baselines to understand how different design decisions impact learned behaviors. I agree with the reviewers that the paper presents a wide breadth of results that illustrate key problems with co-optimizing morphologies and controllers.

**Note From Pc:**

if the above contains the word "oral" or "spotlight" please see: "oral" presentation means -> notable-top-5% and "spotlight" means -> notable-top-25%. As stated in our emails, we are disassociating presentation type from AC recommendations

**Summary Of Ac-Reviewer Meeting:**

N/A